# Effect of Moisture Sources on the Isotopic Composition of Precipitation in Northwest China

Yanlong Kong [1,2,3,*], Ke Wang [1,2,3], Sheng Pan [3,4], Yaqian Ren [1,2,3] and Weizun Zhang [1,5]

1    Key Laboratory of Shale Gas and Geoengineering, Institute of Geology and Geophysics, Chinese Academy of Sciences, Beijing 100029, China
2    Innovation Academy for Earth Science, Chinese Academy of Sciences, Beijing 100029, China
3    College of Earth and Planetary Sciences, University of Chinese Academy of Sciences, Beijing 100049, China
4    Key Laboratory of Continental Collision and Plateau Uplift, Institute of Tibetan Plateau Research, Chinese Academy of Sciences, Beijing 100101, China
5    College of Geoscience and Surveying Engineering, China University of Mining and Technology, Beijing 100083, China
*    Correspondence: ylkong@mail.iggcas.ac.cn

**Abstract:** Stable isotopes ($^{18}O/^{16}O$ and $^2H/^1H$) are fingerprints of water molecules and thus can be used to gain insight on water circulation. Especially, the factors controlling the isotopic composition of precipitation should be identified because they act as baseline determinants of the isotopic variations of surface water and groundwater. Here, using the HYSPLIT model, we attribute observed isotope variations to different moisture sources and characterize the isotopic composition of meteoric precipitation in Northwest China. Results show that the westerlies play a dominant role across the region throughout the year, while other moisture sources only affect some parts of the region during a specific season, i.e., Arctic airflow only affects the Altay Mountains as far as the Middle Tianshan Mountains; the East Asia Monsoon only affects the region east of 100° E longitude during the summer; and summer rainfall of local origin may contribute to the precipitation budget of basin areas. Given the different moisture sources across Northwest China, a local meteoric water line (NWMWL) of $\delta^2H = 6.8\delta^{18}O - 1.6$ is observed. Our findings not only can provide valuable insights into the mechanism of precipitation isotope fractionation in Northwest China but also can contribute to a better understanding of regional climate and hydrological studies.

**Keywords:** moisture sources; stable isotopes; precipitation; HYSPLIT; Northwest China



## 1. Introduction

Arid and semi-arid regions account for about one third of global land area, of which the water shortage affects the lives of billions of people [1–4]. The increasing population growth would bring more prominent conflict on the water resources supply and demand in the arid and semi-arid regions [5–7]. In the future, the water resources in arid and semi-arid regions will become increasingly severe. Northwest China is a typical arid region (Figure 1) [8,9]. In recent years, Northwest China has shown significant warming changes. Over the past 50 years, the average temperature in Xinjiang has increased by 1 °C [10–12], which is higher than the global average increase (0.74 °C) over the past 100 years according to the International Panel on Climate Change (IPCC) [13–15]. As a result of climate change, the runoff flowing out of the mountainous region, which can represent the total water resources in the arid zone, has shown increasing trend with different rates. For example, the runoff of the Urumqi and Kumarak rivers in the north and south of Tianshan in Central Asia has increased by 10.0% and 38.7%, respectively, around 1990 over the past 50 years [16,17]. In order to assess the current situation of water resources, predict future changes in water resources, and improve water resources management measures, a systematic understanding

of precipitation hydrological sources and atmospheric water–surface water–groundwater recharge transformation is required.

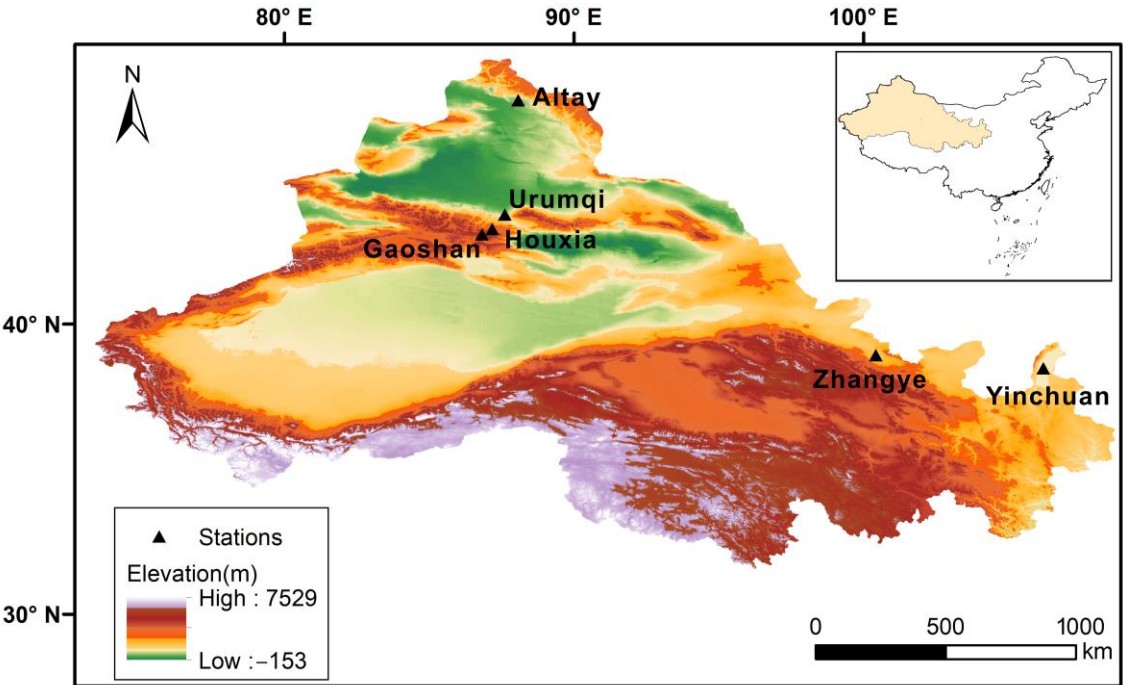

**Figure 1.** Study area and selected stations of precipitation isotopes monitoring across Northwest China.

The Lagrangian Hybrid Single Particle Orbit Model (HYSPLIT), jointly developed by the National Oceanic and Atmospheric Administration (NOAA) Air Resources Laboratory and the Australian Bureau of Meteorology, is able to calculate and analyze the trajectory of moisture transport based on reanalysis data [18,19]. Moisture source regions can be identified according to the HYSPLIT results. However, there might remain some uncertainties for the trajectory of moisture transport because of the simplification of meteorological processes including mixing with surrounding air [20,21]. Stable isotopes of precipitation are also useful tools for analyzing and understanding the processes of moisture transport [22–24]. Meanwhile, numerous studies have shown that moisture sources are important factors controlling the global distribution of precipitation isotopes [25–29]. Jouzel et al. [30] have presented the link between a precipitation isotope and its oceanic origin. Kong et al. [25] have analyzed the seasonal and spatial distribution of precipitation isotopes across the whole China considering the moisture sources. Cai and Tian [31] have attributed the postmonsoon $^{18}$O depletion of a precipitation isotope to moisture transport. Furthermore, the derived parameter deuterium excess (d-excess = $\delta^2$H − 8$\delta^{18}$O) of meteoric precipitation has been proved to be very powerful in tracing moisture sources [32,33]. Over the past decades, a combination of a HYSPLIT model and stable isotopic data has become an effective method to discern the precipitation moisture sources and transport [22,34,35]. Strong et al. [36] used HYSPLIT combined with the distribution of the deuterium isotope to investigate the water-vapor mixing sources at the bottom of the troposphere. Xu et al. [37] discussed the relationship between moisture source regions and $\delta^{18}$O of precipitation in the Namucuo basin of the Tibetan Plateau based on HYSPLIT.

Possible moisture sources in Northwest China include the westerlies, Arctic air masses, the monsoon moisture from west Pacific Ocean, and the local recycled moisture [17,38,39]. However, the extent of these moisture sources and their impact on local precipitation are still controversial [38,39]. Investigation of the relationship between moisture sources and the stable isotopes of precipitation across Northwest China may help to resolve this uncertainty [16,17]. Here, we combine the HYSPLIT model and precipitation isotopes to ascertain the moisture sources of meteoric precipitation across Northwest China.

## 2. Materials and Methods

### 2.1. Data Sources

Stable isotope data, temperature, and precipitation amount at the Urumqi, Zhangye, and Yinchuan stations were obtained from the IAEA Global Network of Isotopes in Precipitation (Figure 1) [40]. The data for Gaoshan Station and Houxia Station are from Pang et al. [41] and Kong et al. [17], which also provide detailed descriptions of the two stations as well as the sampling and analysis procedures. The data for Altay come from Tian et al. [38], where all the data details can be found. A summary of the data used in this study can be found in Table 1.

**Table 1.** Summary of precipitation isotope records at different stations in Northwest China.

| Stations | | Urumqi | Gaoshan | Houxia | Altay | Zhangye | Yinchuan |
|---|---|---|---|---|---|---|---|
| **Annual** | $\delta^{18}O$/‰ | −10.6 | −9.2 | −9.0 | −13.4 | −6.1 | −6.8 |
| | $\delta^2H$/‰ | −71.8 | −63.6 | −63.6 | −97.4 | −40.8 | −43.5 |
| | d-excess/‰ | 12.8 | 9.9 | 8.2 | 9.5 | 7.6 | 11.2 |
| **Summer** | $\delta^{18}O$/‰ | −6.3 | −6.6 | −6.2 | −7.5 | −4.1 | −7.0 |
| | $\delta^2H$/‰ | −42.0 | −45.8 | −43.5 | −52.0 | −28.6 | −49.7 |
| | d-excess/‰ | 8.5 | 7.4 | 6.3 | 7.7 | 3.8 | 6.3 |
| **Winter** | $\delta^{18}O$/‰ | −20.2 | −19.8 | −18.2 | −22.2 | −18.4 | −14.8 |
| | $\delta^2H$/‰ | −141.0 | −138.8 | −128.6 | −168.0 | −123.6 | −103.0 |
| | d-excess/‰ | 20.7 | 19.9 | 17.2 | 9.4 | 23.4 | 15.7 |
| **Maximum** | $\delta^{18}O$/‰ | 1.8 | −6.0 | −4.6 | −5.3 | 0.9 | 3.9 |
| | $\delta^2H$/‰ | −8.9 | −40.5 | −32.7 | −40.2 | −4.3 | 5.1 |
| | d-excess/‰ | 54.8 | 33.5 | 23.7 | 15.5 | 79.0 | 24.2 |
| **Minimum** | $\delta^{18}O$/‰ | −28.0 | −20.8 | −22.2 | −24.2 | −28.5 | −20.0 |
| | $\delta^2H$/‰ | −204.5 | −143.1 | −159.6 | −185.3 | −191.4 | −147.7 |
| | d-excess/‰ | −44.5 | −1.3 | −7.4 | 2.2 | −25.3 | −25.8 |
| **Observation Period** | | 1986–1992, 1996–1998, 2001–2003 | 2003–2004 | 2003–2004 | 1998–2001 | 1986–1992, 1996–1998, 2001–2003 | 1988–1992, 1999–2000 |
| **Source** | | GNIP | Pang et al. [41] | Pang et al. [41] | Tian et al. [38] | GNIP | GNIP |

### 2.2. HYSPLIT Model

To identify the moisture sources of precipitation over Northwest China, the HYSPLIT model was used to simulate the trajectory of moisture transport in the region. Moisture trajectories were mapped for 72 h before reaching the destination. The modeling level height of 1500 m above ground was chosen to show the moisture trajectory because 0–2000 m above ground is the key moisture transport pathway [42]. For the HYSPLIT model results, blue lines were used to represent July 2003 (summer), and red lines indicate January 2004 (winter). We determined the modeling period to be winter and summer based on the seasonal characteristic of precipitation isotopes in Northwest China.

### 2.3. Isotope Data Analysis

All the isotope data used in this work were expressed as permil (‰) difference relative to the VSMOW (Vienna Standard Mean Ocean Water). To obtain the monthly mean value of the precipitation isotope, we took the amount-weighted mean value of all the isotope data within the same month at a certain station.

$$\delta M = \frac{\sum P_i \delta M_i}{\sum P_i} \tag{1}$$

where $\delta M$ is the monthly mean value of the precipitation isotope from all the isotope records, $P_i$ is the precipitation amount during the $i$th month, and $\delta M_i$ is the precipitation isotope data during the $i$th month.

The weighted mean values of the precipitation isotope at both the annual and seasonal scales are following similar calculations with the monthly weighted mean value of precipitation isotopes.

## 3. Results

### 3.1. Moisture Trajectories

Based on the HYSPLIT model results, we found that the moisture trajectory of Gaoshan and Houxia is quite similar to that of Urumqi (Figure 2). Apparently, the westerlies dominate the climate in Northwest China, and the moisture trajectory at the stations over Northwest China bolsters the seasonal shift of the westerlies. From west to east (Gaoshan–Yinchuan), moistures from the west are mixed with more and more moisture from the north during winter, while in summer, almost all the moisture is from the west at Gaoshan and Houxia stations. Monsoon emerges and accounts a considerable proportion in the moisture at the Zhangye station.

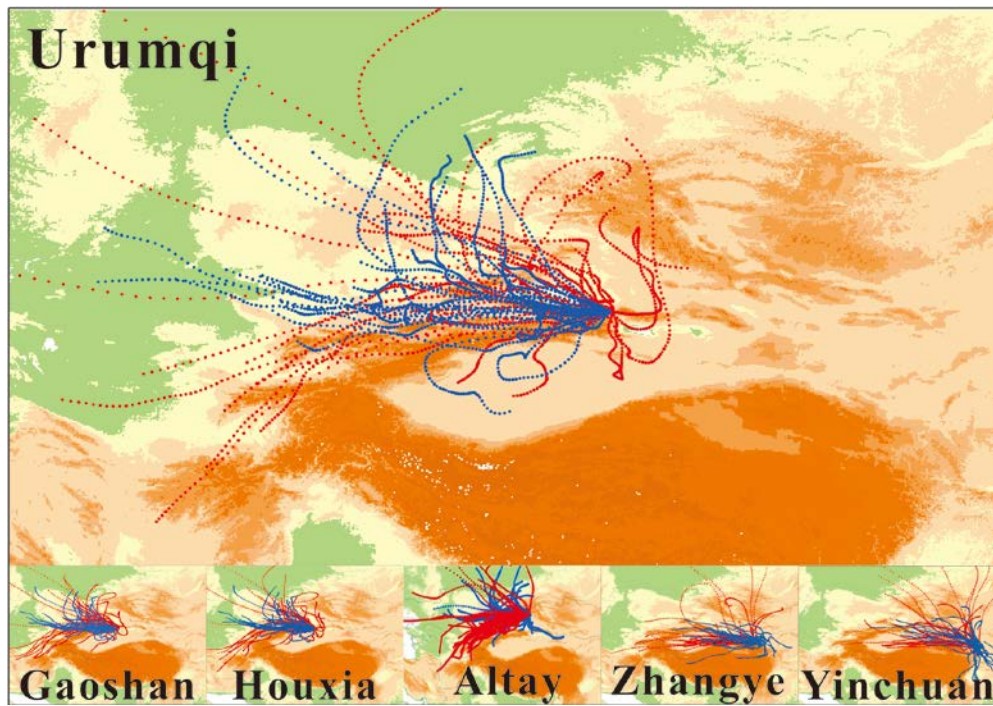

**Figure 2.** Moisture trajectories at selected stations of Urumqi, Gaoshan, Houxia, Altay, Zhangye, and Yinchuan in Northwest China: red lines represent January (winter); blue lines represent July (summer).

### 3.2. Isotopic Compositions of Precipitation

Generally, seasonal variations of precipitation $\delta^{18}O$ and $\delta^2H$ at the 6 stations were similar, reflecting enriched signals in summer and more depleted signals during winter (Table 1 and Figure 3). The enriched isotope signal was usually observed in summer with the maximum value of precipitation $\delta^{18}O$ and $\delta^2H$ to be 1.8‰ and −8.9‰ at the Urumqi station, −6.0‰ and −40.5‰ at the Gaoshan station, −4.6‰ and −32.7‰ at the Houxia station, −5.3‰ and −40.2‰ at the Altay station, 0.9‰ and −4.3‰ at the Zhangye station, and 3.9‰ and 5.1‰ at the Yinchuan station, respectively. During winter, the precipitation isotope was observed to be more depleted than other seasons with the minimum value of precipitation $\delta^{18}O$ and $\delta^2H$ to be −28.0‰ and −204.5‰ at the Urumqi station, −20.8‰ and −143.1‰ at the Gaoshan station, −22.2‰ and −159.6‰ at the Houxia station, −24.2‰ and −185.3‰ at the Altay station, −28.5‰ and −191.4‰ at the Zhangye station, and −20.0‰ and −147.7‰ at the Yinchuan station, respectively. Except for the Altay station, the precipitation d-excess at all the other stations shows seasonal variations with lower values in summer and higher values in winter. The range of precipitation d-excess at the Altay station was much smaller than that at the other stations (Table 1 and Figure 4). The large range of precipitation isotope values and the small range of precipitation d-excess at the Altay station indicate the effect of different moisture sources on the isotope variations and will be discussed in detail in the following sections.

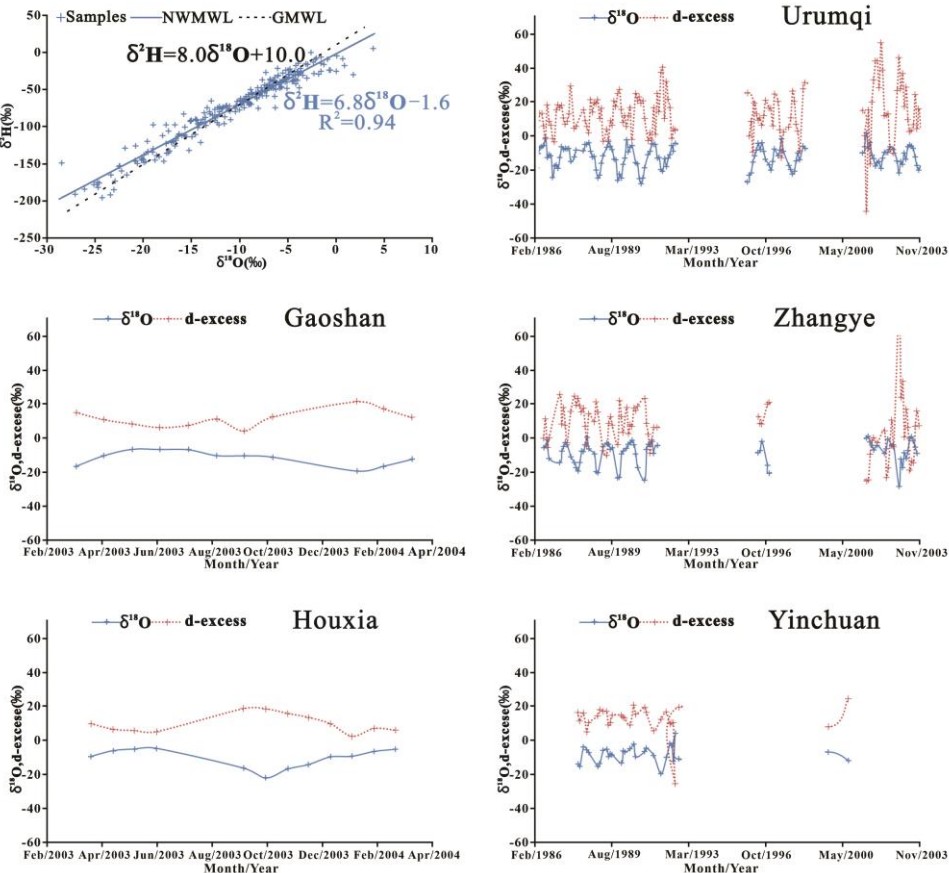

**Figure 3.** Local meteoric water line calculated for Northwest China from this study (**top left**) and variations of precipitation $\delta^{18}$O and d-excess time record for selected stations in Northwest China.

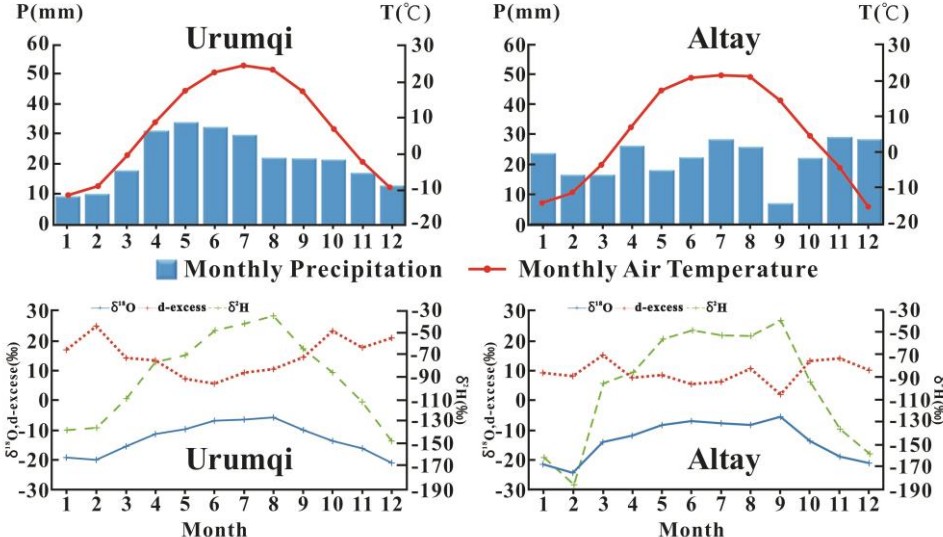

**Figure 4.** Seasonal variations of precipitation amount and temperature at Urumqi and Altay stations, and seasonal variations of $\delta^{18}$O, $\delta^2$H, and d-excess at Urumqi and Altay stations.

*3.3. The Westerlies*

The westerlies are the prevailing winds that deliver moisture derived from the North Atlantic Ocean to Northwest China. In Figure 2, it is observed that most of the blue (summer) and red (winter) lines are from the west, which represents the moisture derived from the westerlies. Thus, the moisture mainly derived from the westerlies is delivered to the region during both winter and summer. Figure 5 illustrates that precipitation in Northwest

China shows a consistent seasonal distribution of more precipitation during the summer season and less precipitation during the winter season. However, both the geographic distribution and d-excess show opposite seasonal characteristics. Summers are characterized by higher $^2$H/$^1$H and $^{18}$O/$^{16}$O ratios and lower d-excess values, whereas precipitation is depleted in $^2$H and $^{18}$O and has higher d-excess values in winter (Figures 3 and 4). Such a distinctive characteristic of the westerlies moisture results from the seasonal swing of the westerlies moisture. In summer, the North Atlantic air is moist, and the d-excess is low. However, the red lines of winter moisture tracks in Figure 2 lie to both the south and north of blue lines of summer moisture tracks, denoting the shift of the westerlies from summer to winter. During winter, the moisture from the Atlantic Ocean mixes the recycled moisture derived from the Caspian Sea, Aral Sea, and other local areas to produce an increase in d-excess values [17,38]. This observation is consistent with the finding of Kreutz et al. [43] based on ice core stable isotopic data from Tianshan Mountains and the finding of Tian et al. [38] regarding the source of meteoric precipitation in Western China.

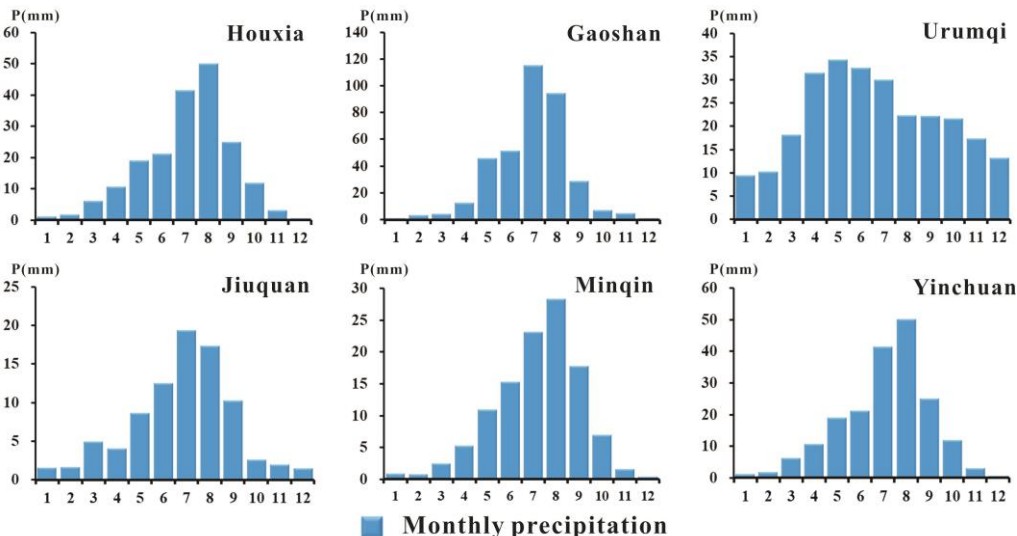

**Figure 5.** Plots of multi-annual average precipitation amount at selected stations in Northwest China.

### 3.4. The Arctic Moisture

The Arctic moisture has some influences on the isotopic composition of meteoric precipitation in the Altay region of northern Xinjiang province, but the extent of the Arctic moisture is not clear. Figure 4 shows that the seasonal variation of temperature at the Urumqi and Altay stations is similar, but the seasonal variation of the precipitation amount is quite different. The precipitation amount is large during summer but low in winter at the Urumqi station, whereas no seasonal difference in the amount of precipitation is observed at Altay station records. At the Urumqi station, summer rainfall is characterized by low d-excess values and winter by high values. This contrasts with the situation at the Altay station, where d-excess values exhibit little annual variation (Figure 4). In summer, the d-excess values at the Altay and Urumqi stations are similar, with all values between 5 and 10‰; while in winter, the d-excess value at the Altay station is below 15‰, and it fluctuates between 15 and 25‰ at the Urumqi Station. This illustrates that the impact of the Arctic moisture on the isotopic composition of precipitation is very weak at the Urumqi station, but it is more significant at the Altay station. Such isotopic difference between the Tianshan and Altay mountains was also found by Li et al. [44]; however, we can further draw the conclusion from our work that the Arctic moisture almost has no influence on precipitation isotopes in Northwest China during summer, and its influence only extends as far as the Altay mountains and the north on the Tianshan mountains of Northwest China in winter.

### 3.5. The Monsoon

The monsoon includes the Indian monsoon derived from the south and the East Asian monsoon from the southeast. Tian et al. [38] pointed out from the research on precipitation isotopes in the Qinghai-Tibet Plateau that the northward boundary of the Indian Ocean monsoon is 35° N latitude, but controversy still exists about the westward boundary of the East Asian monsoon. It is traditionally thought that the East Asian summer monsoon can only reach 100° E longitude [39,45–47], whereas Xu et al. [48] claimed on the basis of tree-ring stable isotope variations that the East Asian monsoon might have influenced the west of 100° E during the period of 1883–1975. Figure 2 shows that the subtropical ocean moisture has no impact on Northwest China in winter. However, in summer, moisture from the east is present over the Yinchuan region, extends part way into the Zhangye region, and is not present over the Urumqi region. Additionally, it is known that the isotopic composition of precipitation affected by the East Asian monsoon always exhibits a precipitation amount effect [39] that is not observed in Northwest China by this study. Therefore, the East Asian monsoon, if present, has only a very minor effect on the isotopic composition of meteoric precipitation in Northwest China.

### 3.6. Local Recycled Moisture

Local recycled moisture does not contribute much to the precipitation in Northwest China. Taking the Urumqi station as an example, it is about 8% of the annual precipitation [17]. However, even this low contribution can alter the d-excess value of precipitation because, characteristically, recycled moisture always has large d-excess [17,49]. Both $^2H/^1H$ and $^{18}O/^{16}O$ ratios of precipitation and d-excess values are variable at different stations across Northwest China, including at Xinjiang, Gansu, and Ningxia provinces, thus documenting a small local moisture effect on the stable isotopic composition of precipitation.

## 4. Discussion

### 4.1. The Northwest Regional Meteoric Water Line (NWMWL)

Meteoric water lines are important reference lines in the hydrological and hydrogeological research by using stable isotopes ($\delta^2H$ and $\delta^{18}O$). The global meteoric water line (GMWL) is known as $\delta^2H = 8.0\delta^{18}O + 10.0$ [50]. The local meteoric water line (LMWL) is usually used in regional studies due to the differences in geographical characteristics and moisture sources around the world. However, the problem we often face is that precipitation sampling stations are very rare, and, thus, it is difficult to obtain the proper LMWL. Ordinarily, the LMWL is determined based on the precipitation isotope data at a local station (such as GNIP stations). If the moisture sources are different, the meteoric water lines are different at different stations. Nevertheless, a regional meteoric water line can be made when the moisture sources are the same across a whole region. According to the results of the analysis above, the northwest region is mainly dominated by the westerlies, and the moisture sources are basically the same at different stations. Therefore, we use the precipitation isotopes at the Gaoshan, Houxia, Urumqi, Zhangye, and Yinchuan stations to make the regional meteoric water line as the northwest region meteoric water line (NWMWL): $\delta^2H = 6.8\delta^{18}O - 1.6$ (Figure 3). The slope of the NWMWL is 6.8, which is less than the slope of the global meteoric water line (8.0), reflecting the sub-cloud evaporation process in the arid area [51–54]. The NWMWL found in this work is similar with previous studies in Northwest China, but there is a little difference that is caused by the differences of the database [25,55].

### 4.2. Implications for the Effect of Climate Change on Water Cycle

From a holistic view of the earth system, precipitation, surface water, and groundwater all occupy important positions on the exchange of materials and energy during geological processes. General circulation models (GCMs) are tools for large-scale simulations of natural atmospheric cycles and can predict future climate patterns. Precipitation isotopes as well as d-excess can serve as calibration tools for general circulation models (GCMs) to qualify

the complex water-vapor mixing processes [56–59]. Modern atmospheric precipitation isotope monitoring data have been shown to be a direct record of climate change [60–62]. Variations of precipitation isotopes in continental regions can reflect local climatic conditions, especially for moisture source regions [30,35,63]. In Northwest China, there are more and more publications reporting that a warmer and more humid climate is coming, while the cause is still not quite clear. Therefore, the application of precipitation isotopes together with GCMs might help in this regard. Furthermore, the utilization of precipitation isotopes has the ability to unify the interrelationships between the precipitation, surface water, and groundwater, which will advance the investigation on the composition and movement within water systems, as well as the exchange of fluxes, solutes, and energy occurring at the boundaries of water systems. Therefore, by investigating the moisture sources on precipitation isotopes in Northwest China, it can furnish a scientific foundation for an exact evaluation of the effect of climate change on the water cycle in the arid Northwest China.

## 5. Conclusions

By analyzing the precipitation isotopes in Northwest China and employing the HYS-PLIT model, the following conclusions are drawn:

(1) The dominant moisture source affecting Northwest China is the westerlies. Other less important sources include the Arctic moisture, monsoon, and local recycled moisture. These moisture sources have a variable impact regionally. During different seasons, the Arctic moisture affects the Altay mountains throughout the year and the northern part of Tianshan mountains in winter, the east Asian monsoon impacts the region east of 100° E longitude in summer, and local recycled moisture affects the basin region in the summer and autumn seasons.

(2) The local meteoric water line for the Northwest China region (NWMWL: $\delta^2 H = 6.8\delta^{18}O - 1.6$) is a representative line on the feature of meteoric sources across the region and, therefore, can be used as the reference line for hydrological circulation studies in Northwest China.

**Author Contributions:** Conceptualization, Y.K.; methodology, Y.K.; validation, K.W., S.P. and Y.R.; formal analysis, Y.K.; investigation, K.W., S.P. and Y.R.; resources, Y.K.; data curation, S.P. and Y.R.; writing—original draft preparation, Y.K. and K.W.; writing—review and editing, Y.K., K.W., S.P., Y.R. and W.Z.; visualization, K.W., S.P., Y.R. and W.Z.; supervision, K.W. and W.Z.; project administration, Y.K.; funding acquisition, Y.K. All authors have read and agreed to the published version of the manuscript.

**Funding:** This research was funded by the Third Xinjiang Scientific Expedition Program (grant number 2022xjkk1304) and the National Natural Science Foundation of China (grant numbers U1703122 and 91647101).

**Data Availability Statement:** The data are available upon request from the corresponding author.

**Conflicts of Interest:** The authors declare no conflict of interest.

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
