# Peer review of "Effect of Moisture Sources on the Isotopic Composition of Precipitation in Northwest China"

_water, doi:10.3390/w15081584_

Round 1

Reviewer 1 Report

see the attachment

Author Response

The paper presents analysis of stable isotopes (18O/16O and 2H/1H) in precipitation in Northwest China. The observed variations were attributed to different sources of moisture, among which westerlies are the dominant one. Local meteoric water line, NWMWL, was also determined. However, the main question remains the use of HYSPLIT model. The way the results are presented could have been done by simple analysis of the data, without the model. More about use of HYSPLIT model should be described – I do not see a word of the model results. The authors state that (line 55 and line 62) the HYSPLIT model is a method to discern the effect of precipitation moisture sources.

In spite of this question, I think that the results are worth publishing. Precipitation of NW China are seldom the subject of the research, and this paper gives the insight to the topics, especially the difference of Altai station in comparison to other stations.

The paper is rather well written, conclusions are consistent with the presented details.

The paper deserves to be published in Climate.

Response: Thanks for your valuable comments. For the HYSPLIT model, we have added the descriptions of the model running settings like running time and modelling height in the Methods section as well as the HYSPLIT model results in the Results section. The newly added sections have provided detail information to explain the HYSPLIT model.

For the HYSPLIT model results, there might remain some uncertainties about the trajectory of moisture transport for the simplification of meteorological processes like mixing with surrounding air [Noone and Sturm, 2010; Dutsch et al., 2018]. To discern the effect of precipitation moisture sources, precipitation isotopes can serve as powerful tools because they could record information about the moisture sources and transport as well as other atmospheric processes (Tian et al., 2007; Kong et al., 2019). We have rephrased our writing to make it clearer to the readers in the Introduction section.

Noone, D., & Sturm, C. (2010). Comprehensive dynamical models of global and regional water isotope distributions. Isoscapes: Understanding movement, pattern, and process on Earth through isotope mapping, 195-219.

Dütsch, M., Pfahl, S., Meyer, M., & Wernli, H. (2018). Lagrangian process attribution of isotopic variations in near-surface water vapour in a 30-year regional climate simulation over Europe. Atmospheric Chemistry and Physics, 18(3), 1653-1669.

Tian, L., Yao, T., MacClune, K., White, J. W. C., Schilla, A., Vaughn, B., ... & Ichiyanagi, K. (2007). Stable isotopic variations in west China: A consideration of moisture sources. Journal of Geophysical Research: Atmospheres, 112(D10).

Kong, Y., Wang, K., Li, J., & Pang, Z. (2019). Stable isotopes of precipitation in China: A consideration of moisture sources. Water, 11(6), 1239.

There are some deficiencies that should also be resolved, especially Figure 3.

Response: We have revised the manuscript following the reviewers’ comments, details can be found from the newly revised version.

Abstract, line 17: semicolon (:) is not needed here, just coma (,). If the semicolon stays, then you have to re-arrange the rest of the sentence.

Response: Thanks. We have replaced the (:) with (,).

Line 27: ”rate” is too much – “The increasing population growth would bring … “

Response: Revised.

Line 33: 100a → 100 years

Response: Modified.

Line 38 – 41: unclear sentence; perhaps change to “In order to assess …, to predict future …. and to improve water …”

Response: We have corrected this sentence following your comments. And the new sentence goes like “In order to assess the current situation of water resources, to predict future changes in wa-ter resources and to improve water resources management measures, a systematic under-standing of precipitation hydrological sources and atmospheric water - surface water - groundwater recharge transformation is required.”

Line 44: “Laboratory” should be written together

Response: Revised.

Line 52: deuterium isotope – singular

Response: Revised.

Figure 1: Urumqi is missing in the caption. Are all the trajectories only for January (red) and July

(blue)? If so, please, write “lines”. What are the black lines at 4 station?

Response: Thanks for the reviewer’s valuable comments. Urumqi has been added in the caption. Yes, all the blue lines represent January while red lines represent June. We have used ‘lines’ instead of ‘line’ in the revised manuscript. We redraw this Figure to keep all the color to be same for all the stations, which can be found in the revised Figure 2.

Line 61 and 73: Pang et al. [27] and Tian et al [24] – be consistent in writing references, with or without “.” – check also the rest of the paper

Response: Revised.

Line 82: in Urumqi maximum is from April to July

Response: Revised.

Line 85: is “D” d-excess or 2H? 18 in superscript

Response: Revised.

Line 86: Figure 3

Response: Revised.

Figure 2 (and all the rest of the figures) – frame of the figures is not necessary

Response: We have removed all the unnecessary frame of the figures.

Figure 3: first panel: write equation with only two decimal places; y = 6.81 x - 1.60, R2 = 0.94; the rest of panels – write “d-excess” correctly at all of them, x-axes – use consistent writing of month/year, yaxes – plot 0 ‰, and then the range of (negative) values for δ18O and (positive) values for d-excess; letters for stations are not equal; Yinchuan – check the values, seem not all values are in increasing order?

Response: We have replotted this figure following your comments following your suggestions, details about the revisions are as follows: 1. We have changed the equations with only two decimal places; 2. “d-excess” has been corrected in all the subplot; 3. We have used the same x-axes lables ‘month/year’ format in all the subfigures and y-axes has also corrected to make a uniform look with 0‰ inside; 4. Letters for the stations has kept the same in the new figure; 5. We delete the lines during the gap years without observation data to avoid misunderstanding.

Please find the above revisions in the new figure.

Lines 110 – 112: confusing sentences - some re-arrangement is needed. Perhaps you may delete the sentence beginning with “However…”

Response: Revised. We have deleted this sentence.

Line 113: Figure 4

Response: Revised.

Figure 4. replace “D-excess” with “d-excess” on y-axes

Response: Modified.

Line 145: D/H → 2H/1H, or δ2H and δ18O

Response: Revised. We have replaced 2H/1H with D/H.

Line 151: hydrological and hydrogeological – lower case letters

Response: Revised.

Line 165: Figure 3

Response: Revised.

Line 169: occupy

Response: Revised.

Line 172: can be served → can serve

Response: Revised.

Line 183; section number 5

Response: Revised.

References are appropriate. Please, use the same font size for all the references

Response: Revised. And all the font size of the references are the same.

Line 224: spaces in front of and after “-“

Response: Revised.

Line 230: Hydrology

Response: Revised.

Lines 233 – 234: Development … China … its ... Environment

Response: Revised.

Line 256: pages are missing or paper number

Response: Revised. Pages and paper number have been added.

Line 257: δ18O

Response: Revised.

Line 260, 268, 277, 297, 298, 306: pages are missing or paper number

Response: Revised. Pages and paper number have been added.

Line 269: Wang P. K: → Wang, P. K.

Response: Revised.

Line 273: the ‘86/87 and ‘91/92 (spaces in front of ‘)

Response: Revised.

Line 276: δ18O

Response: Revised.

Line 280: delta 18O → δ18O

Response: Revised.

Line 281; lower case letters (except first letter in a word)

Response: Modified.

Line 283: earth sciences → Earth Sciences

Response: Revised.

Reviewer 2 Report

Dear Authors,

Unfortunately, I cannot recommend that this paper be published in its present form.

The paper’s title suggest that it is a moisture sources on the isotopic composition of precipitation study; however, the isotopic composition of precipitation is probably the least discussed of the material in the paper.

There is also insufficient detail on the:

1.      Introduction : there are insufficient current references regarding the composition of stable isotopes, moisture sources and the use of the HYSPLIT method.

The purpose of the article is not well highlighted, but only the application of the HYSPLIT model.

Moreover, a map with the study area and the analyzed stations should have appeared in this section.

2.      Data and methods.

The manuscript contains so many fundamental misunderstandings about the data, methods, and model that I cannot recommend it for publication.  In this section, data regarding the analyzed period should have appeared. According to the graphs, there are clear differences between the analysis periods at the analyzed stations. Moreover, there are no data specified by the HYSPLIT model such as:  trajectory direction, vertical motion, period analysed, total run time, level height. The model was run for days with significant precipitation? Modelling at three levels is the most common situation in these kinds of publications (recently).

3.      Results and discussion.

The results only describe the moisture source, without having a statistic of the moisture sources (there are no graphic or table in the manuscript). Moreover, there are no discussions about the stable isotopes in precipitation and d-excess at the analysed stations, like  minimums, maximums and averages. The connection between d-excess and the data from the moisture sources is not highlighted and represented.

The manuscript does not explain why only two months (July and January) were used for the moisture trajectories.

At the very least, the authors need to rewrite the paper more clearly in order to reflect the contents.

Author Response

Unfortunately, I cannot recommend that this paper be published in its present form.

Response: Thanks for the reviewer’s effort to giving comments on our work and we have revised the whole manuscript following the reviewer’s comments to improve the quality of our paper. And we hope the revised manuscript can meet the expectation of the reviewer.

The paper’s title suggest that it is a moisture sources on the isotopic composition of precipitation study; however, the isotopic composition of precipitation is probably the least discussed of the material in the paper.

Response: Thanks for your nice comments. We have added more descriptions of precipitation isotopes both in the result and discussion section (especially the sub-section 2.1, 2.3 and 3.2). Besides, a summary of the isotope data at the 6 stations has been listed in Table 1. Revisions can be found in the newly submitted manuscript.

There is also insufficient detail on the:

  1. Introduction: there are insufficient current references regarding the composition of stable isotopes, moisture sources and the use of the HYSPLIT method.

The purpose of the article is not well highlighted, but only the application of the HYSPLIT model.

Moreover, a map with the study area and the analyzed stations should have appeared in this section.

Response: Thanks for your valuable suggestions. We have added and summarized the works on the composition of stable isotopes, moisture sources and the use of the HYSPLIT method to make it clearer to the readers. The main purpose of this work is to astern the effect of the moisture sources on the precipitation with the combination of HYSPLIT model and precipitation isotope records, and we rephrased our sentences to make it more clearly. We have enhanced our work with more descriptions of precipitations isotopes as well as the HYSPLIT model results to make it more clearly. The revisions can be found in the newly submitted manuscript.

We have added a new figure to show the location of study area and selected stations, which can be found in the new Figure 1.

  1. Data and methods.

The manuscript contains so many fundamental misunderstandings about the data, methods, and model that I cannot recommend it for publication.  In this section, data regarding the analyzed period should have appeared. According to the graphs, there are clear differences between the analysis periods at the analyzed stations. Moreover, there are no data specified by the HYSPLIT model such as:  trajectory direction, vertical motion, period analysed, total run time, level height. The model was run for days with significant precipitation? Modelling at three levels is the most common situation in these kinds of publications (recently).

Response: Thanks for your suggestions. We made major revisions about this section. A table to summarize the precipitation isotope record information the analyzed period has been newly added. And the isotope data analysis section has been implemented in the Method section to explain our ways to process the isotope data. For the HYSPLIT model, we added a new section to describe the detail information about this method and results has also been explained in the Results section. The long-term average and trend of stable isotopes are used to make the comparison to keep the logic analysis. While for the modelling level, we have model the trajectory of moisture at three levels including 500m, 1500m and 2500m and the height level of 1500m was finally chosen as the representative height. (Please see the new section 2)

  1. Results and discussion.

The results only describe the moisture source, without having a statistic of the moisture sources (there are no graphic or table in the manuscript). Moreover, there are no discussions about the stable isotopes in precipitation and d-excess at the analysed stations, like minimums, maximums and averages. The connection between d-excess and the data from the moisture sources is not highlighted and represented.

The manuscript does not explain why only two months (July and January) were used for the moisture trajectories.

Response: Thanks for your comments. In the results section, we have newly implemented two sections to describe the HYSPLIT model as well as the isotopic composition of precipitation over the stations including the characteristic of precipitation δ18O, δ2H as well as d-excess. And the connections between d-excess and stable isotopes are also used to compare with the moisture trajectories (Please see the subsection 3.3, 3.4, 3.5 and 3.6).

For the HYSPLIT model, we select two seasons, winter and summer, to trace the trajectory of moisture transport corresponding to the seasonal variation of precipitation isotopes. Modelling results of July represent summer and that of January stands for winter, which follows the tradition of most climate studies in China such as the following references:

Xing, J., Wang, J., Mathur, R., Wang, S., Sarwar, G., Pleim, J., ... & Hao, J. (2017). Impacts of aerosol direct effects on tropospheric ozone through changes in atmospheric dynamics and photolysis rates. Atmospheric Chemistry and Physics, 17(16), 9869-9883.

Zhang, N., Gao, Z., Wang, X., & Chen, Y. (2010). Modeling the impact of urbanization on the local and regional climate in Yangtze River Delta, China. Theoretical and applied climatology, 102, 331-342.

Feng, F., Li, Z., Zhang, M., Jin, S., & Dong, Z. (2013). Deuterium and oxygen 18 in precipitation and atmospheric moisture in the upper Urumqi River Basin, eastern Tianshan Mountains. Environmental Earth Sciences, 68, 1199-1209.

At the very least, the authors need to rewrite the paper more clearly in order to reflect the contents.

Response: Thanks for your great effort to review our work. We carefully revised our work following your valuable comments and the revisions can be found in the newly submitted manuscript.

Reviewer 3 Report

This work evaluates the effect of moisture sources on the precipitation isotopes in Northwest China. Besides applying HYSPLIT model to investigate the moisture sources, the isotopic composition and deuterium-excess of precipitation has been used to help ascertain the affected arid regions of different moisture sources in Northwest China. The findings of this work provide important information to understand the mechanism of precipitation distribution in arid regions and meet the scope of Climate. Therefore, I recommend to accept the paper after a minor revision.

Some specific comments for the authors:

1.       At the end of ABSTRACT section, the implications of the findings in this work including on the use of stable isotopes in the hydrological studies should be addressed to reveal the significance of this work.

2.       In the ‘introduction’ section, many new works on the effect of moisture sources on stable isotopes should be cited, and their contribution should be summarized.

3.       In section 2 ‘Data sources’, the ‘Altai’ should be ‘Altay’, check the whole manuscript to make sure they are the same.

4.       In section 2, a table to summarize the data is recommended to put in the manuscript. This could help the readers (especially the international readers) to have a basic concept on the stable isotope values in the Northwest regions.

5.       Before the section of ‘Results’, there should be 1 section of ‘method’ to the Hysplit model and isotope data analysis.

6.       What does it mean by the ‘airflow system’ in the first sentence of section 3.1, please rephrase.

7.       In section 3.3, replace ‘Indian ocean moisture’ with ‘Indian Monsoon moisture’.

8.       In section 4.1, ‘The local meteoric water lines (LMWL) are usually used in regional studies due to the differences in geographical characteristics and moisture sources around the world.’ In this sentence, it should be ‘The local meteoric water line (LMWL) is usually used …’

9.       The title of section 4.2, I suggest it to change as ‘Implications for the effect of climate change on water cycle’

10.   Conclusions should be the 5th section.

Author Response

This work evaluates the effect of moisture sources on the precipitation isotopes in Northwest China. Besides applying HYSPLIT model to investigate the moisture sources, the isotopic composition and deuterium-excess of precipitation has been used to help ascertain the affected arid regions of different moisture sources in Northwest China. The findings of this work provide important information to understand the mechanism of precipitation distribution in arid regions and meet the scope of Climate. Therefore, I recommend to accept the paper after a minor revision.

Response: Thanks for the reviewer’s effort to review our work and valuable suggestions to improve the paper. We have carefully revised the manuscript following your comments. Revisions can be found in the newly submitted manuscript.

Some specific comments for the authors:

  1. At the end of ABSTRACT section, the implications of the findings in this work including on the use of stable isotopes in the hydrological studies should be addressed to reveal the significance of this work.

Response: Thanks for your nice suggestion. We added the significance of our work in the ABSTRACT section, which goes like ’Our findings can not only provide valuable insights into the mechanism of moisture transport in Northwest China, thereby contributing to a better understanding of regional climate and hydrological studies.’

  1. In the ‘introduction’ section, many new works on the effect of moisture sources on stable isotopes should be cited, and their contribution should be summarized.

Response: Thanks for your comments. We re-summarized the works on the effect of moisture sources and implemented new references on this topic, which can be found in the revised manuscript.

  1. In section 2 ‘Data sources’, the ‘Altai’ should be ‘Altay’, check the whole manuscript to make sure they are the same.

Response: Changed.

  1. In section 2, a table to summarize the data is recommended to put in the manuscript. This could help the readers (especially the international readers) to have a basic concept on the stable isotope values in the Northwest regions.

Response: Thanks for your suggestion. We implemented a new table to summarize the isotope records. Details can be found in the newly submitted manuscript.

  1. Before the section of ‘Results’, there should be 1 section of ‘method’ to the Hysplit model and isotope data analysis.

Response: Thanks for your valuable suggestion. The two sections has been added to describe the method of HYSPLIT model as well as the isotope data analysis, which can be found in the revised manuscript.

  1. What does it mean by the ‘airflow system’ in the first sentence of section 3.1, please rephrase.

Response: Thanks for your comments. We have replaced the ‘airflow system’ with ‘prevailing winds’ in the manuscript.  

  1. In section 3.3, replace ‘Indian ocean moisture’ with ‘Indian Monsoon moisture’.

Response: Corrected.

  1. In section 4.1, ‘The local meteoric water lines (LMWL) are usually used in regional studies due to the differences in geographical characteristics and moisture sources around the world.’ In this sentence, it should be ‘The local meteoric water line (LMWL) is usually used …’

Response: Revised accordingly.

  1. The title of section 4.2, I suggest it to change as ‘Implications for the effect of climate change on water cycle’

Response: Thanks for your suggestion and we have revised the title of section 4.2 as ‘Implications for the effect of climate change on water cycle’.

  1. Conclusions should be the 5th section.

Response: Corrected.

Round 2

Reviewer 1 Report

There are minor corrections to be made. The most concern is data in Table 1 for Altai station.

Here are the comments:

line 58 and 59: has --> have

table 1, raw "source" - Tian et al. [38]; add space between "et al." and [41]; is it true that average winter d-excess value for Altai is "-9.37"?

line 100: 2000m --> 2000 m

line 105:  "delta permil" replace with "permil (‰) difference"

lines 109 -111: (1) to the right; δM and ??? should be unified

line 120: combine sentences "... China and the ..."

line 123: combine sentence "... winter, while in summer almost all .."; comma not needed after "summer"

table 1 and lines 130 -148 (and possibly some other): please, do not give δ2H values with two decimal places

Author Response

There are minor corrections to be made. The most concern is data in Table 1 for Altai station.

Response: Thanks for the reviewer’s valuable efforts and suggestions on our work. We have double-checked our data and make sure all the data in this work is correct especially for Table 1. The revisions can be found from the following point-to-point response and newly submitted manuscript.

Here are the comments:

line 58 and 59: has --> have

Response: Corrected.

table 1, raw "source" - Tian et al. [38]; add space between "et al." and [41]; is it true that average winter d-excess value for Altai is "-9.37"?

Response: Revised. The formatting suggestions have been revised accordingly. The average winter d-excess value for Altai is 9.37 rather than -9.37 and we have corrected it in Table 1. Besides, all the data in Table 1 has been re-checked by us to make sure the table without mistakes.

line 100: 2000m --> 2000 m

Response: Corrected.

line 105:  "delta permil" replace with "permil (‰) difference"

Response: Corrected.

lines 109 -111: (1) to the right; δM and ??? should be unified

Response: Modified.

line 120: combine sentences "... China and the ..."

Response: Revised accordingly.

line 123: combine sentence "... winter, while in summer almost all .."; comma not needed after "summer"

Response: Revised accordingly.

table 1 and lines 130 -148 (and possibly some other): please, do not give δ2H values with two decimal places

Response: Thanks to the reviewer’s nice suggestion. We have revised the isotope data to keep with one decimal place for both δ2H and δ18O in Table 1 and the whole manuscript, which can be found in the newly submitted manuscript.

Reviewer 2 Report

Dear authors,

Thank you for submitting the revised manuscript. A few more suggestions to insert on the new version.

Page 5, Figure 2: Please indicate in percentages the moisture sources identifies.

Page 6, Figure 4. Please add the GMWL line, in order to explain the NLMWL from lines 240-242.

Page 8, line 278 - 280: Please add the percentage that explain the moisture sources.

Author Response

Dear authors,

Thank you for submitting the revised manuscript. A few more suggestions to insert on the new version.

Response: Thanks for the reviewer’s efforts to reviewing our work and providing valuable suggestions to improve the quality of the paper. We have fully considered your comments and revised our work as the following response.

Page 5, Figure 2: Please indicate in percentages the moisture sources identifies.

Response: Thanks for the reviewer’s comments. The HYSPLIT results were shown in Figure 2 to indicate the wind directions at different stations during both winter and summer seasons. The results could indicate where the wind comes from during the past period. However, it’s hard to accurately calculate the moisture percentages of different source regions from only the trajectories by HYSPLIT model because it presents the wind direction but not vapor in the air, as we have illustrated in the Introduction section. That’s why we combined the precipitation isotopes with HYSPLIT model to infer the possible moisture source region.

Page 6, Figure 4. Please add the GMWL line, in order to explain the NLMWL from lines 240-242.

Response: Thanks for the reviewer’s nice suggestion. We have added the GMWL as well as its equation in Figure 4. Details can be found in the newly submitted Figure 4.

Page 8, line 278 - 280: Please add the percentage that explain the moisture sources.

Response: As the moisture sources are qualitatively determined by stable isotopes and HYSPLIT model, we used the qualitative description such as ‘dominance’ instead of percentage. Please also see the response above.